# Pre-Treatment Neutrophil-to-Lymphocyte and Platelet-to-Lymphocyte Ratios as Predictors of Occult Cervical Metastasis in Clinically Negative Neck Supraglottic and Glottic Cancer

**DOI:** 10.3390/jpm11121252

**Published:** 2021-11-25

**Authors:** Giovanni Salzano, Francesco Perri, Fabio Maglitto, Giulia Togo, Gianluca Renato De Fazio, Michela Apolito, Federica Calabria, Claudia Laface, Luigi Angelo Vaira, Umberto Committeri, Mario Balia, Ettore Pavone, Corrado Aversa, Francesco Antonio Salzano, Vincenzo Abbate, Alessandro Ottaiano, Marco Cascella, Mariachiara Santorsola, Roberta Fusco, Luigi Califano, Franco Ionna

**Affiliations:** 1Maxillo-Facial and ENT Surgery Unit, INT-IRCCS “Fondazione G. Pascale”, 80131 Naples, Italy; giovanni.salzano@istitutotumori.na.it (G.S.); umbertocommitteri@gmail.com (U.C.); e.pavone@istitutotumori.na.it (E.P.); c.aversa@istitutotumori.na.it (C.A.); f.ionna@istitutotumori.na.it (F.I.); 2Head and Neck Medical and Experimental Oncology Unit, INT IRCCS Fondazione Giovanni Pascale, 80131 Naples, Italy; 3Maxillofacial Surgery Unit, Department of Neurosciences, Reproductive and Odontostomatological Sciences, University Federico II, 80131 Naples, Italy; fabio.maglitto@istitutotumori.na.it (F.M.); giulia.togo@gmail.com (G.T.); gianludefazio@gmail.com (G.R.D.F.); michela.apolito@gmail.com (M.A.); calabria.federica@gmail.com (F.C.); clalaface@gmail.com (C.L.); vincenzo.abbate@unina.it (V.A.); luigi.califano@unina.it (L.C.); 4Maxillofacial Surgery Unit, University Hospital of Sassari, 07100 Sassari, Italy; luigi.vaira@gmail.com; 5Maxillofacial Surgery Unit, University of Naples Federico II, Via Sergio Pansini 5, 80131 Naples, Italy; mario.balia@studenti.unicampania.it; 6Department of Medicine, Surgery and Dentistry, “Scuola Medica Salernitana”, University of Salerno, 84081 Baronissi, Italy; frsalzano@unisa.it; 7SSD Innovative Therapies for Abdominal Metastases, Istituto Nazionale Tumori IRCCS Fondazione G. Pascale, 80131 Naples, Italy; a.ottaiano@istitutotumori.na.it (A.O.); mariachiara.santorsola@istitutotumori.na.it (M.S.); 8Division of Anesthesia, Istituto Nazionale Tumori, IRCCS Fondazione G. Pascale, 80131 Naples, Italy; m.cascella@istitutotumori.na.it; 9Oncology Medical Division, Igea SpA, 80127 Naples, Italy; r.fusco@igeamedical.com

**Keywords:** laryngeal carcinoma, neutrophil-to-lymphocyte ratio, platelet-to-lymphocyte ratio, neck metastases, elective neck dissection

## Abstract

*Background.* Among patients with diagnosis of Laryngeal Squamous Cell Carcinoma (LSCC), up to 37.5% of cases may have occult metastasis (OM), and this feature is linked to poor prognosis and high rate of local recurrence. The role of elective neck dissection (END) in clinically negative neck (cN0) LSCC remains controversial. It is of great value to search for low-cost and easily detectable indicators to predict the risk of OM in laryngeal cancer. Recent reports have shown that high values of the neutrophil-to-lymphocyte ratio (NLR) and platelet-to-lymphocyte ratio (PLR) represent a negative prognostic factor in head and neck cancers. The aim of our study has been to investigate the value of pre-treatment NLR and PLR with regard to predicting occult cervical metastasis in cN0 supraglottic and glottic LSCC. Materials and methods. Data of patients affected by LSCC, who had been surgically treated by means of laryngectomy (total, horizontal partial and supracricoid) and END between January 2006 and January 2021, were retrospectively reviewed, using information retrieved from a database dedicated to such procedures in a single tertiary care referral institute. Results. A total of 387 patients were treated for LSCC at our Institute from 2006 to 2021, but only 108 of them met the inclusion criteria. The median age at the time of diagnosis was 64 years (range, 39–89 years). All the tumors were treated with a laryngectomy and an END. A total of 27.7% of patients were found positive for neck node metastasis (the pN+ group), while 78/108 (72.3%) patients were found to be negative for the presence of neck metastasis (the pN0 group). High values of NLR, but not PLR, significantly correlated with the probability of OM, and according to the iterative algorithm of Newton–Raphson, an NLR value of 2.26 corresponds to a probability of OM of 20%. *Conclusion.* Our analysis revealed a statistical correlation between high NLR pre-treatment values and positive neck OM in patients with LSCC.

## 1. Introduction

The presence of cervical lymph node metastasis in Laryngeal Squamous Cell Carcinoma (LSCC) is notorious in negatively affecting the survival of patients [1]. The status of the cervical lymph nodes is acknowledged to be the most important prognostic factor in this kind of malignancy, and a diagnostic delay in the neck may result in treatment failure [2,3]. Among patients with diagnosis of LSCC, up to 37.5% of cases may have occult metastasis (OM) that is undetectable using current imaging techniques, such as magnetic resonance imaging (MRI), computed tomography (CT), ultrasound and positron-emission tomography (PET) [4]. 

The OM percentage may be so high because, especially in the supraglottic region, there is a robust lymphatic network that generally includes neck levels II and III. In early-stage glottic cancer, the percentage of cervical metastasis is <10%, although these tumors may hide OM at a range of between 10% and 35% [5]. For this reason, an appropriate management of the cervical lymph nodes is mandatory for disease control. Nowadays, there are four main therapeutic alternatives: neck irradiation (NI), watchful waiting (WW), sentinel node biopsy (SND) and elective neck dissection (END). It is currently widely accepted that END is justified if the risk of metastasis is higher than 15–20% [2]. 

The role of END in clinically negative neck (cN0) LSCC remains controversial since, as widely described in the literature, this approach is not free from complications. Consequently, variations in the type and extent of surgical dissection have been discussed. At present, the traditional surgical management of cN0 laryngeal cancer is a bilateral neck dissection level II-IV in tumors classified as T3-T4 and in cases of centrally located or unilateral T1-T2 supraglottic tumors reaching the midline [6,7,8]. The limited involvement at level IV has raised doubts in respect of the cost–benefit ratio in the opinion of some authors [9,10]. 

The grade of disease, the cartilage invasion, the size of the tumor and the perineural and vascular invasion are current markers described to calculate the risk of OM in cN0 LSCC. 

It is of great value to search for low-cost and easily detectable indicators to predict the risk of OM in laryngeal cancer. It is now widely demonstrated that an inflammatory response plays a critical role in the promotion and progression of the tumor, potentially influencing survival outcomes [11,12]. The neutrophil-to-lymphocyte ratio (NLR) and platelet-to-lymphocyte ratio (PLR) are the markers reflecting systemic inflammation and may be easily calculated from a blood sample [13,14,15]. Recent reports have shown that high values of the NLR and PLR represent a negative prognostic factor in head and neck cancers [13,16,17]. 

The aim of our study has been to investigate the value of pre-treatment NLR and PLR with regard to predicting occult cervical metastasis in cN0 supraglottic and glottic LSCC in order to obtain a new predictive parameter that can assist the surgeon in the determination of the type and extent of surgical neck dissection required.

## 2. Materials and Methods

All patients affected by LSCC, who had been surgically treated by means of laryngectomy (total, horizontal partial and supracricoid) and END between January 2006 and January 2021, were retrospectively reviewed using information retrieved from a database dedicated to such procedures in a single tertiary care referral institute. The study was performed in compliance with the Helsinki Declaration and with policies approved by the local board of ethics. 

The inclusion criteria were as follows: (1) a histology-proven case of LSCC; (2) primary LSCC that had not been treated previously; (3) LSCC with no regional or distant metastasis detectable at presentation, clinically and/or radiographically (Neck Ultrasound, CT and/or MRI); (4) no previous cancers at any other sites; (5) no radiotherapy or chemotherapy in the clinical history; (6) no clinical conditions that might affect the NLR or PLR (e.g., an inflammatory, autoimmune, acute or chronic infectious disease, hematological disorder, a history of corticosteroid therapy or chronic renal insufficiency); (7) a complete record of pre-treatment hematological variables; and (8) treatment with a laryngectomy (total, horizontal partial or supracricoid) and an END at level II-IV (monolateral or bilateral according to the location of the primary tumor). The treatment and follow-up pre-treatment work-up consisted of a complete physical examination, a fiber optic nasopharyngoscopy, routine blood counts, liver function tests, a neck ultrasound and a total body CT scan with contrast enhancement. The local extension of the disease was estimated with a multiplanar CT scan and contrast-enhanced MRI in all cases. All the tumors were retrospectively staged according to the TNM classification (eighth edition) [18]. From a histological viewpoint, the laryngeal squamous cell carcinomas (LSCC) were categorized according to the World Health Organization/International Agency for Research on Cancer classification of human tumors, which is accepted and used worldwide [19]. Our sample was divided into two groups based on the presence (pN+ group) or absence (pN0 group) of metastatic neck nodes proven on END histopathology. Written informed consent was obtained from all the patients prior to any diagnostic or therapeutic procedure. All the patients have provided written informed consent for the publication of their data in anonymous form.

The study was conducted according to the guidelines of the Declaration of Helsinki. Informed consent was obtained from all the subjects involved in the study. The patients in this project consented to use of all photographs and illustrations and data for the purposes of educational content. The signed consent forms are on file at the maxillofacial and ENT department of the National Cancer Institute “G. Pascale” of Naples, Naples, Italy, containing personally identifiable patient information and signatures. These data are available upon specific request.

## 3. Statistical Analysis

The NLR was calculated as the absolute neutrophil count divided by the absolute lymphocyte count, whereas the PLR was calculated as the platelet count divided by the total lymphocyte count. 

A logistic regression model was performed to estimate the probability π (x) of cervical node metastasis from a study of the NLR and PLR levels in the blood tests. The analysis was performed using the iterative algorithm of Newton–Raphson:(1)$$\pi \;\left(x\right)=\frac{\exp\left(\mathrm{Intercept}+\beta \cdot x\right)}{1+\exp\left(\mathrm{Intercept}+\beta \cdot x\right)}$$
where π (x) indicates the probability of finding NLR or PLR in the cN0 for a given NLR or PLR (the x value). The Intercept parameter and β were estimated using the Curve Fitting Toolbox of Matlab R2007a (MathWorks, Natick, MA, USA). 

Linear regression was used to assess the correlation between NLR or PLR with respect to presence of node metastasis.

Linear regression model was used to assess the accuracy of NLR or PLR in the identification of metastatic nodes considering the receiver operating characteristic (ROC) analysis and the Youden index to identify the optimal cut-off value. Considering the optimal cut-off value, we reported sensitivity, specificity, positive predictive value (PPV) and negative predictive value (NPV).

The non-parametric Kruskal–Wallis test was performed to evaluate any statistically significant differences in the NLR or PLR values between the group of patients with pN0 and the group of patients with pN+. The Chi square test was used to evaluate the correlation between the T and N stages in the group with pN+. A *p* value of <0.05 was considered significant.

The statistical analysis was performed using the Statistics Toolbox of Matlab R2007a (MathWorks, Natick, MA, USA).

## 4. Results

A total of 387 patients were treated for LSCC at our Institute from 2006 to 2021. Of these cases, a total of 279 patients were excluded because they did not meet the inclusion criteria: 47 because they had come to our attention with regional metastasis; 14 because they had presented with distant metastasis; 18 because they had previously had cancers at other sites (colorectal, breast, prostate, lung or kidney cancer); 161 because they had undergone a cordectomy for glottic cancer without an END; 14 because they had been affected by a recurrence of LSCC when referred to our Institute; and 25 due to comorbidities that might affect the NLR or PLR value. 

The 108 patients (108/379, 28.5%) who respected the inclusion criteria for the present analysis were enrolled in the study. The median age at the time of diagnosis was 64 years (range, 39–89 years). All the tumors were treated with a laryngectomy and an END. Thirty-two patients underwent a horizontal laryngectomy/supracricoid laryngectomy, and seventy-six patients underwent a total laryngectomy. In five cases, a unilateral neck dissection was performed for primary lesions particularly distant form the midline. One hundred three patients underwent an elective bilateral neck dissection level II-IV. Therefore, in total, the neck was dissected in 211 cases. A free margin resection of the primary LSCC was obtained in all cases. 

From a pathological point of view, 30/108 (27.7%) patients were found positive for neck node metastasis (the pN+ group), while 78/108 (72.3%) patients were found to be negative for the presence of neck metastasis (the pN0 group). Among the 103 patients who underwent a bilateral neck dissection, the rate of contralateral neck metastasis was 1.9% (2/103). Levels II and III of the neck were the most frequently affected areas, with only 2 patients out of 108 developing OM at level IV, a prevalence of 1.85% (2/108). 

With regard to the T classification, 8 patients were T1, 51 were staged as T2, 44 were T3 and 5 were T4a. Perineural invasion was assessed by histological evaluation in 23 patients out of 108 (21.2%), and lymphovascular invasion was assessed in 34 out of 108 patients (31.4%). 

Table 1 reports the distribution of the N stage in the population with respect to the T stage. The *p* value at the Chi square test was 0.06, thus demonstrating that there was no correlation between the T and N stage.

Table 2 reports the distribution of the N stage in the population with respect to the grading. The *p* value at the Chi square test was 0.80, thus demonstrating that there was no correlation between the grading and N stage. 

The average NLR level was 2.15 (1.13–5.64) for the PN0 group and 2.87(1.24–6.13) for the PN+ group (*p* < 0.01 at the Kruskal–Wallis test, Figure 1). The average PLR level was 119.25 (46.64–274.68) for the PN0 group and 136.81 (82.53–272.22) for the PN+ group (*p* < 0.05 at the Kruskal–Wallis test, Figure 1).

Figure 2 shows a graphical representation of our statistical model for the NLR.

The overlapping of PLR values in the PN0 and PN+ groups did not allow us to identify any Intercept and β values in the logistic regression model for the PLR. 

In Figure 3, the mathematical relationship between the NLR and the probability of the occurrence of occult cervical metastasis is illustrated. For every 10 units, an increase in the probability of the occurrence of occult cervical metastasis in relation to the NLR values was calculated using the designed model. An NLR value of 2.26 corresponds to a probability of OM of 20%.

Table 3 shows the Intercept and β values estimated using the logistic regression model for the NLR and PLR models.

Table 4 reports the multivariate analysis considering linear regression of all clinicopathologic variables. Among the considered variables, exclusively NLR and T stage influenced the N stage significantly.

Table 5 reports the linear regression model of NRL, PLR and of all clinico-pathologic variables with respective accuracy in the identification of metastatic nodes. Figure 4 reports the ROC curves. 

## 5. Discussion

LSCCs are the most common head and neck tumors, accounting for 1–2% of malignant tumors. Metastatic lymph node involvement is an important prognostic factor which influences the surgical approach and the subsequent therapeutic plan. Therefore, the need to identify markers of lymph node metastasis is crucial [1,2].

LSCC accounts for a small fraction of new cancer cases each year (about 0.8%). Nevertheless, it is a pathology with severe psychological impact on the patient, due to the possible phonatory alteration following surgery and the potential sequelae of adjuvant treatments. Among the numerous risk factors for LSCC, cigarette smoking is the principal determinant, followed by human papillomavirus infection [20], laryngopharyngeal reflux, environmental exposure and alcohol abuse. As the number of smokers has reduced, the incidence of LSCC has decreased by about 2.4% in the last 10 years. However, the five-year survival has remained almost unchanged [21]. 

The presence of lymph node metastasis in LSCC can be explained in terms of the large lymphatic network of the anatomical district, which represents a negative survival index. Early-stage LSCCs have a low incidence of cervical lymph node metastasis (0–10%), the incidence increasing up to 35% in advanced-stage LSCC, with a prevalent localization at the lymph node levels II and III [5,22].

The most widely used approach for the treatment of high-stage LSSC without any radiological evidence of lymph node involvement is surgical intervention by means of a bilateral END [23,24]. Historically, END has been recommended in cases where there is a risk of OM exceeding 15% to 20% in order to achieve an optimal balance between the risk of recurrence and the risk of overtreatment [25]. 

END is a valid tool for oncological staging, providing useful information for post-surgical radiotherapy, especially in clinically negative advanced-stage primary tumors.

Sharbel et al. suggested that the rate of OM is greater in patients with advanced LSCC. It rests at 13% in patients with T1–T2 tumors and tends to increase with the progression of the T classification, reaching 25% in patients with T3–T4 tumors. Therefore, T3–T4 patients should undergo END despite having a cN0 [26]. Bilateral selective neck dissection is recommended for all LSCC patients with a symmetrical lesion or bulky carcinoma involving both sides of the larynx, while the ipsilateral neck dissection of levels II–IV is the surgical approach recommended in cases of LSCC being limited to one side of the larynx and with respect to small T2 tumors [27].

Until recently, the surgical approach consisted of a modified neck dissection type III (functional neck dissection) with the removal of levels I, II, III, IV and V and the preservation of the sternocleidomastoid muscle, the internal jugular vein, the spinal accessory nerve and the submandibular gland. Currently, however, a selective neck dissection (lateral neck dissection) of levels II to IV is preferred because levels I and V are rarely the site of lymph node metastasis. Ferlito et al. [9] have demonstrated that level II B dissection may be unnecessary in patients with LSCC, thereby reducing the risk of dysfunction of the spinal nerve, thus preferring the dissection of the sublevel of IIA–III only. The requirement to remove level IV is debated in literature, since most patients with level IV metastasis already have level II/III positive lymph nodes. Therefore, the dissection of level IV might not be necessary in patients with a cN0 neck [2].

In accordance with the NCCN Guidelines 2018 [28], our surgical approach for LSCC includes the END of levels II–IV, except in the case of small unilateral or low-stage LSCCs, in respect of which, as recommended in the literature, we prefer to perform an ipsilateral neck dissection. 

In agreement with Zhang et al. [27], only 2/103 patients in our study presented contralateral metastases after bilateral END. The adequacy of a bilateral END in patients of Group b according to Zhang (cN0 laryngeal cancer classified as T3–T4 and in cases of centrally located or unilateral T1–T2 supraglottic tumors reaching the midline) remains to be evaluated because it can represent an overtreatment.

Radical neck dissection is commonly considered an overtreatment because it includes the removal of level I and V nodes [29]. 

Ozdek et al. suggest that tumor stage, invasive tumor margins and lymphovascular invasion represent the main markers of metastasis in LSCC (13), or, in any case, they correlate with the metastatic potential of the tumor. Other tumor features related to an increased risk of OM are poorly differentiated (G2–G3) tumors [30,31,32,33,34,35].

Pre-treatment hematological markers have emerged as prognostic factors for several cancers. Moreover, combining multiple prognostic factors may give more precise information on any lymph node involvement, thus enabling an optimization of the treatment to improve survival outcomes. Indeed, this approach should prove useful in assisting surgeons in the identification of patients at a higher risk of lymph node metastasis and in the determination of the type of neck dissection preferred.

The correlation between inflammation and oncogenesis has already been studied, with hematological biomarkers playing a relevant role in tracking tumor progression, thereby influencing survival outcomes. The NLR [36,37] and PLR [17,38] are biomarkers of systemic inflammation which can be evaluated on a pre-operative peripheral blood sample and are therefore easily assessable without any additional cost, due to the fact that blood tests are routinely performed in every patient. 

The negative prognostic value of these biomarkers has already been highlighted in the literature in relation to tumors of the head and neck district, such as nasopharyngeal cancer [37,38], oral squamous carcinoma [12] and paranasal sinus cancer [13]. They have also been evaluated with respect to tumors of other anatomical regions [39,40,41,42,43,44]. Several studies have shown how the number of granulocyte-monocyte progenitors in circulation is negatively related to progression-free survival, because their increase is induced by cancer secreted cytokines (such as Interleukine-6), with the leukocytes releasing pro-tumorigenic cytokines, stimulating signals such as the factor k pathway [45,46]. According to the literature, tumor-associated thrombocytosis is a poor prognostic factor in several tumor types, and high platelet counts have been correlated with an increased risk of metastasis [47], considering that high platelet levels may induce an extravasation of cancer cells and increase micro-vessel permeability and angiogenesis [48].

Kum et al. showed that patients with pre-cancerous laryngeal lesions had a significantly lower NLR than LSCC patients, assuming that this biomarker can be used to differentiate malignant from benign laryngeal lesions [49]. Abbate et al. [12] demonstrated a statistically significant relationship between a pre-operative NLR > 2.93 and the risk of neck OM in patients with a squamous cell carcinoma of the oral tongue. Ozturk et al. showed that high PLR and NLR levels in early-stage tongue cancers are related to a high rate of local recurrence, without affecting the overall survival or progression-free survival [15].

To the best of our knowledge, this is the first study evaluating the correlation between high levels of the NLR and PLR in patients with LSCC and the risk of occult lymph node metastasis. Our results have shown that the NLR is useful for the identification of patients with LSCC at the highest risk of neck nodal OM, thereby guiding the surgeon in the determination of the most effective type of END in relation to the cN0. Our analysis revealed a statistical correlation between high NLR pre-treatment values and positive neck OM in the population under study, showing how an NLR value of 2.26 corresponds to a 20% probability of lymph node OM. The NLR increases in a way directly proportional to the probability of developing OM, an NLR value of 4.09 corresponding to a risk of the occurrence of OM of approximately 90%.

However, low accuracy was obtained considering NRL or PLR alone in the identification of metastatic nodes (accuracy 60% and 70%, respectively), while a good accuracy of 80% was obtained considering the linear regression model of all clinico-pathologic variables (NLR, PLR, T stage, grading, perineural invasion and lymphovascular invasion).

Due to the small study population, it was not possible to identify a statistically significant correlation between high pre-operative PLR values and the risk of occult lymph node metastasis. Therefore, further studies on a larger series will be needed to define a statistically relevant correlation.

The mathematical relationship between NLR and the presence of OM is slightly exponential, with a maximum 0.41 differential increase in the NLR values calculated between 80% (3.68) to 90% (4.09) in relation to the probability of finding OM. The linear increase observed in our models makes it difficult to identify a cut-off value that can adequately guide the therapeutic choice. A solution could be provided by means of an identification of the inflection point from our distribution. From a statistical point of view, the inflection point of a distribution indicates the point where a concave curve becomes convex, changing course and sign. In our distribution, this point corresponds to an NLR value of 2.26 and a probability of finding neck OM of 20%. Beyond this value, small increases in the NLR determine an exponential increase in the risk of the occurrence of OM nodes. The latter corresponds to the value above, which means the probability of finding metastasis in a cN0 increases exponentially. In particular, an increase of 0.27 in the NLR value corresponds to a 10% increased risk of occult lymph node metastasis.

Based on this mathematical model, we have identified the threshold value probability of finding OM, in addition to providing a recommendation for the performance of an END. These preliminary results offer clinicians and surgeons an easy, economic and obtainable tool with which to stratify patients based on the risk of the occurrence of metastatic lymph nodes and thus to determine which cases an END could be recommended for. Further studies on a larger series will be needed to confirm the results obtained and to define more precisely the NLR value appropriate for a recommendation of an END in LSCC cN0 patients.

## Figures and Tables

**Figure 1 jpm-11-01252-f001:**
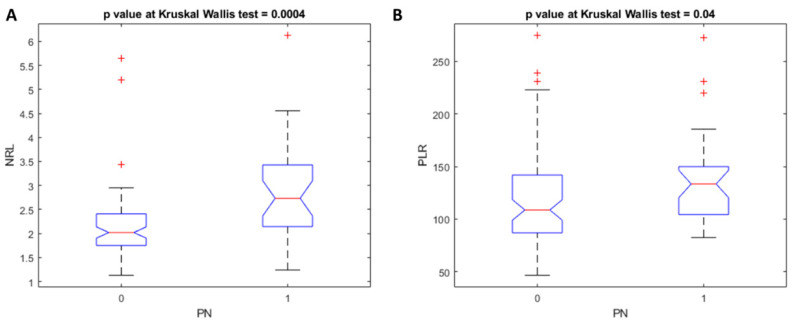
Boxplot of the NLR (**A**) and PLR (**B**).

**Figure 2 jpm-11-01252-f002:**
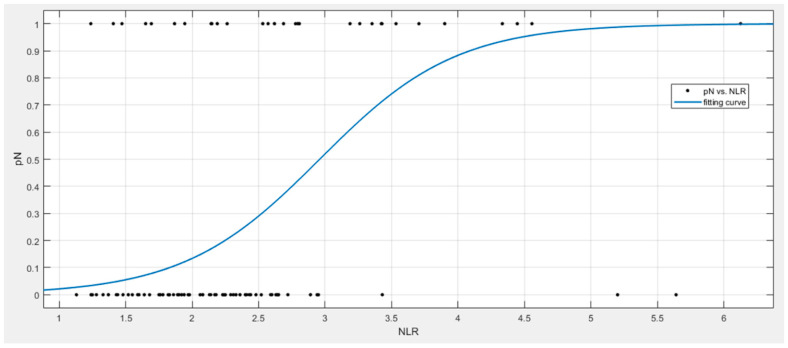
Graphical representation of our statistical model of the NLR.

**Figure 3 jpm-11-01252-f003:**
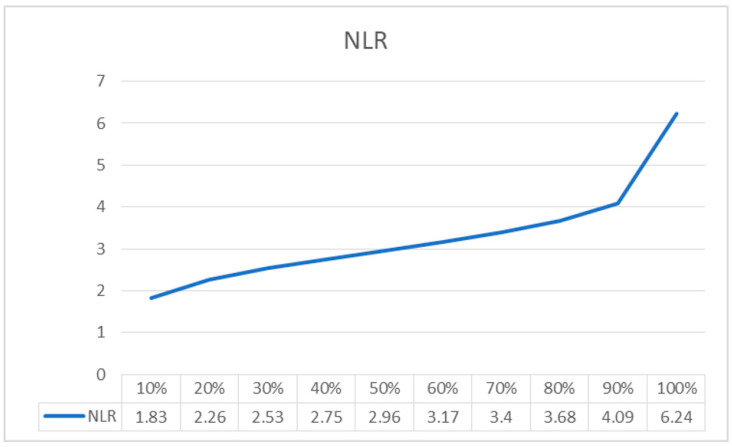
Curve showing the mathematical relationship between the NLR and the probability of the occurrence of occult cervical metastasis.

**Figure 4 jpm-11-01252-f004:**
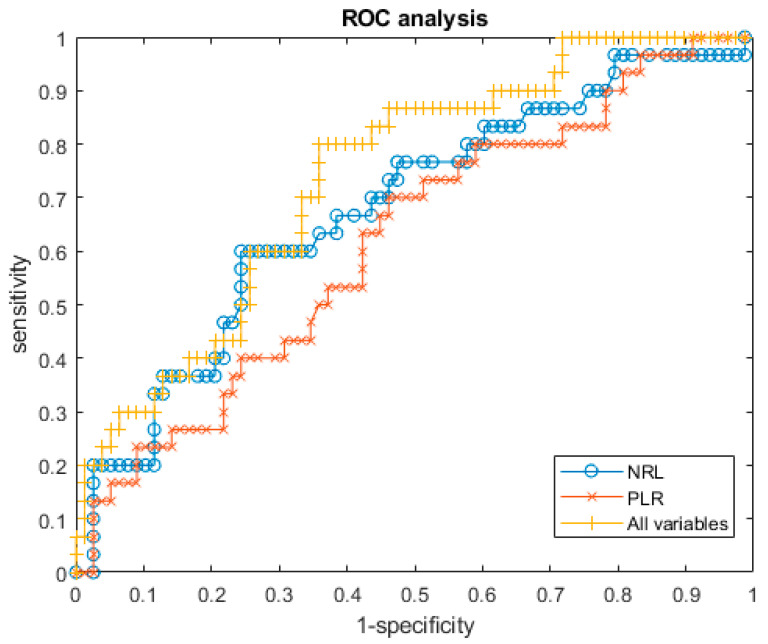
ROC analysis for NRL, PLR and multivariate regression model.

**Table 1 jpm-11-01252-t001:** Distribution of the N stage in the population with respect to the T stage.

	N0	N1	N2a	N2b	N2c
T1	8				
T2	40	4	5	1	1
T3	27	7	3	6	1
T4	3	1	1		

**Table 2 jpm-11-01252-t002:** Distribution of the N stage in the population with respect to the grading.

	N0	N1	N2a	N2b	N2c
G1	1				
G2	29	3	4	1	
G3	48	9	5	6	2

**Table 3 jpm-11-01252-t003:** Estimate of the Intercept and β values for the NLR and PLR models.

Factor	Estimate	95% Confident Interval
Intercept_NLR_	−5.76	−8.63; −2.89
β_NLR_	1.95	0.85; 3.04

**Table 4 jpm-11-01252-t004:** Multivariate analysis considering linear regression of all clinico-pathologic variables.

	Coefficients	*p* Value	*p* Value at Kruskal–Wallis Test
Intercept	−0.27	0.30	0.01
NLR	0.05	0.04	
PLR	0.00	0.93	
T stage	0.14	0.02	
Grading	0.01	0.90	
Perineural invasion	0.14	0.19	
Lymphovascular invasion	−0.09	0.31	

**Table 5 jpm-11-01252-t005:** Linear regression analysis of NRL, PLR and of all clinico-pathologic variables.

NRL Linear Regression Model	Coefficients	*p* Value	AUC	Accuracy	Sensitivity	Specificity	PPV	NPV	Cut-Off
Intercept	0.06	0.482	0.67	0.60	0.76	0.49	0.83	0.71	0.27
NRL	0.07	0.002							
PLR Linear Regression Model	Coefficients	*p* value	AUC	Accuracy	Sensitivity	Specificity	PPV	NPV	Cut-off
Intercept	153.58	2.4 × 10^−20^	0.61	0.70	0.54	0.37	0.82	0.58	4378.12
PLR	34.22	0.18							
Linear regression model of all clinico-pathologic variables			0.73	0.80	0.64	0.46	0.89	0.69	0.26

AUC: area under ROC curve; PPV: positive predictive value; NPV: negative predictive value.

## Data Availability

All data are reported in the manuscript.

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
