# Peer review of "Pre-Treatment Neutrophil-to-Lymphocyte and Platelet-to-Lymphocyte Ratios as Predictors of Occult Cervical Metastasis in Clinically Negative Neck Supraglottic and Glottic Cancer"

_jpm, 2021, doi:10.3390/jpm11121252_

Round 1

Reviewer 1 Report

In this retrospective study, the authors investigate the correlation between the preoperative findings of NLR and PLR and the presence of the occult neck metastases in patients suffering from the glottic or supraglottic laryngeal SCC.

The study is very well designed and the paper is clearly written.

Still, I would recommend some minor supplements in the text:

  1. Raw 196 and 200: At the end of the sentence adequate reference is needed.
  2. In the discussion chapter it is necessary to refer to the extremely small percentage of contralateral neck metastases in this series of patients (2/103) and compare it with the results of similar studies in the literature. Could it be a typical example of an overtreatment?

Author Response

Reviewer 1

In this retrospective study, the authors investigate the correlation between the preoperative findings of NLR and PLR and the presence of the occult neck metastases in patients suffering from the glottic or supraglottic laryngeal SCC.

The study is very well designed and the paper is clearly written.

Still, I would recommend some minor supplements in the text:

  1. Raw 196 and 200: At the end of the sentence adequate reference is needed.

As suggested by reviewer, we updated the bibliography.

  1. In the discussion chapter it is necessary to refer to the extremely small percentage of contralateral neck metastases in this series of patients (2/103) and compare it with the results of similar studies in the literature. Could it be a typical example of an overtreatment?

As suggested by reviewer, we evaluated and compared to literature the percentage of controlateral neck metastases in our patients

Reviewer 2 Report

This interesting article evaluates the association of NLR and PLR with the presence of occult metastases in patients with cN0 laryngeal carcinoma who received elective neck dissection. An association is found between an elevated NLR and the presence of occult metastases, and how the incidence of these occult metastases increases in parallel with the elevation of the NLR.

Overall, the article is original, presents novel information and is well designed and conducted. Obviously, the work requires validation in other series of patients with laryngeal cancer, but the information presented is novel and relevant. However, there are some aspects that I believe could be improved.

-Both the introduction and the discussion indicate that treatment of the cN0 neck in laryngeal cancer consists of bilateral dissection of levels II-IV. But this is only justified in tumors classified as T3-T4 and in cases of centrally located or unilateral T1-T2 supraglottic tumors reaching the midline; moreover, in glottic T1-T2, elective neck dissection is not indicated (Cancers (Basel). 2020 Apr 24;12(4):1059; Head Neck. 2006 Jun;28(6):534-9).

Elsewhere in the discussion these indications are somewhat clarified, but it is better that the text be modified to make the indications for elective neck treatment clear. Moreover, the paper's own results, which show a very low incidence of contralateral occult metastases, argue against the routine indication of bilateral dissection (this is a valuable conclusion that can also be drawn from the paper).

- Only the association of the T classification with metastases is shown, and in my opinion, in the wrong way: instead of showing and analyzing the T classification in pN+ cases, the pN classification should be shown and analyzed according to the T classification. Moreover, a study of the association between clinicopathologic variables (T, G, perineural and vascular invasion) and occult metastases is missing. They should make an analysis of all these variables in relation to N, and make a multivariate analysis with the significant ones.

- In paragraph 3 of the discussion they confuse on several occasions "grade" (which is histologic) with "stage" (which is clinical/pathologic). This should be corrected.

- Also in paragraph 3 of the discussion it is mentioned that "historically" elective dissection is considered indicated when the incidence of occult metastases is greater than 15-20%; this is not a historical whim, it is a threshold based on the study by Weiss et al (Arch Otolaryngol Head Neck Surg. 1994 Jul;120(7):699-702), which should be cited.

- In paragraph 7 of the discussion, extranodal invasion (ENE) is included as a risk factor for occult metastases, but this cannot be such a risk factor, since it is a parameter that is analyzed in the metastases themselves (i.e. it is already known that there are metastases).

- The authors establish a linear relationship between the NLR value and the presence of metastases, and indicate that this precludes establishing a cut-off point, but this optimal cut-off point (the NLR value that would be associated with the highest sensitivity and specificity) could be assessed by ROC curve analysis (or at least attempted).

Author Response

Reviewer 2

This interesting article evaluates the association of NLR and PLR with the presence of occult metastases in patients with cN0 laryngeal carcinoma who received elective neck dissection. An association is found between an elevated NLR and the presence of occult metastases, and how the incidence of these occult metastases increases in parallel with the elevation of the NLR.

Overall, the article is original, presents novel information and is well designed and conducted. Obviously, the work requires validation in other series of patients with laryngeal cancer, but the information presented is novel and relevant. However, there are some aspects that I believe could be improved.

-Both the introduction and the discussion indicate that treatment of the cN0 neck in laryngeal cancer consists of bilateral dissection of levels II-IV. But this is only justified in tumors classified as T3-T4 and in cases of centrally located or unilateral T1-T2 supraglottic tumors reaching the midline; moreover, in glottic T1-T2, elective neck dissection is not indicated (Cancers (Basel). 2020 Apr 24;12(4):1059; Head Neck. 2006 Jun;28(6):534-9).

As suggested by reviewer, we clarified the indication to the bilateral END and updated the bibliography.

Elsewhere in the discussion these indications are somewhat clarified, but it is better that the text be modified to make the indications for elective neck treatment clear. Moreover, the paper's own results, which show a very low incidence of contralateral occult metastases, argue against the routine indication of bilateral dissection (this is a valuable conclusion that can also be drawn from the paper).

As suggested by reviewer , we added in the paragraph 3 of the discussion that the bilateral END can be considered an overtreatment in some laryngeal cancer  (cN0 laryngeal cancer classified as T3-T4 and in cases of centrally located or unilateral T1-T2 supraglottic tumors reaching the midline): we are working on a prospective study to evaluate the bilateral END in these cases of laryngeal cancer.

- Only the association of the T classification with metastases is shown, and in my opinion, in the wrong way: instead of showing and analyzing the T classification in pN+ cases, the pN classification should be shown and analyzed according to the T classification.

As suggested by reviewer, Table 1 was replaced to show pN classification according to the T classification.

Moreover, a study of the association between clinicopathologic variables (T, G, perineural and vascular invasion) and occult metastases is missing. They should make an analysis of all these variables in relation to N, and make a multivariate analysis with the significant ones.

As suggested by reviewer we reported in the Table IV the multivariate analysis considering linear regression of all clinic-pathologic variables.

- In paragraph 3 of the discussion they confuse on several occasions "grade" (which is histologic) with "stage" (which is clinical/pathologic). This should be corrected.

As suggested by reviewer, the text have been corrected

- Also in paragraph 3 of the discussion it is mentioned that "historically" elective dissection is considered indicated when the incidence of occult metastases is greater than 15-20%; this is not a historical whim, it is a threshold based on the study by Weiss et al (Arch Otolaryngol Head Neck Surg. 1994 Jul;120(7):699-702), which should be cited.

As suggested by reviewer, we updated the bibliography.

- In paragraph 7 of the discussion, extranodal invasion (ENE) is included as a risk factor for occult metastases, but this cannot be such a risk factor, since it is a parameter that is analyzed in the metastases themselves (i.e. it is already known that there are metastases).

As suggested by reviewer, the text have been corrected

- The authors establish a linear relationship between the NLR value and the presence of metastases, and indicate that this precludes establishing a cut-off point, but this optimal cut-off point (the NLR value that would be associated with the highest sensitivity and specificity) could be assessed by ROC curve analysis (or at least attempted).

As suggested by reviewer we performed linear regression modelling for single considered variable and for all clinic-pathologic variables with respective roc analysis and accuracy values (Table V and Figure 4).

Round 2

Reviewer 2 Report

The authors have adressed all que issues raised by the reviewer. 

However, the new tables III and IV, and the new figure 4 are not mentioned in the text of the results section. It is mandatory that these new data are described in the results and also discussed. 

Author Response

Dear reviewer, thanks for the suggestion, however in the results section we
already reported the citations of news table and figure as follows:

Table III shows the intercept and β values estimated using the Logistic
Regression model for the NLR and PLR models
Table IV reports the multivariate analysis considering linear regression of
all clinic-pathologic variables. Exclusively NLR and T stage among the
considered variables influenced the N stage significantly.
Table V reports the linear regression model of NRL, PLR and of all
clinic-pathologic variables with respective accuracy in the identification
of metastatic nodes. Figure 4 reports the ROC curves.

Moreover, in the discussion section was added the following paragraph:
However, low accuracy was obtained considering NRL or PLR alone in the
identification of metastatic nodes (accuracy 60% and 70% respectively) while
a good accuracy of 80% was obtained considering the linear regression model
of all clinic-pathologic variables (NLR, PLR, T stage, grading, perineural
invasion and lymphovascular invasion)

Best Regards

Dr Francesco Perri (corresponding author)
